# Phytochemical Characterization, and Antioxidant and Antimicrobial Properties of Agitated Cultures of Three Rue Species: *Ruta chalepensis*, *Ruta corsica*, and *Ruta graveolens*

**DOI:** 10.3390/antiox11030592

**Published:** 2022-03-20

**Authors:** Agnieszka Szewczyk, Andreana Marino, Jessica Molinari, Halina Ekiert, Natalizia Miceli

**Affiliations:** 1Department of Pharmaceutical Botany, Faculty of Pharmacy, Jagiellonian University Medical College, 30-688 Krakow, Poland; halina.ekiert@uj.edu.pl; 2Department of Chemical, Biological, Pharmaceutical and Environmental Sciences, University of Messina, Viale F. Stagno d’Alcontres, 31, 98166 Messina, Italy; anmarino@unime.it (A.M.); jessica.molinari@studenti.unime.it (J.M.); nmiceli@unime.it (N.M.)

**Keywords:** *Ruta* *chalepensis*, *Ruta corsica*, *Ruta graveolens*, in vitro cultures, linear furanocoumarins, furoquinoline alkaloids, antioxidant activity, antimicrobial activity

## Abstract

The in vitro cultures of the following three species of the genus *Ruta* were investigated: *R. chalepensis*, *R. corsica*, and *R. graveolens*. The dynamics of biomass growth and accumulation of secondary metabolites in the 3-, 4-, 5-, 6-, and 7-week growth cycle were analysed. The antioxidant capacity of the methanol extracts obtained from the biomass of the in vitro cultures was also assessed by different in vitro assays: 1,1-diphenyl-2-picrylhydrazil (DPPH), reducing power, and Fe^2+^ chelating activity assays. Moreover, a preliminary screening of the antimicrobial potential of the extracts was performed. The extracts were phytochemically characterized by high-performance liquid chromatography (HPLC), which highlighted the presence of linear furanocoumarins (bergapten, isoimperatorin, isopimpinellin, psoralen, and xanthotoxin) and furoquinoline alkaloids (γ-fagarine, 7-isopentenyloxy-γ-fagarine, and skimmianine). The dominant group of compounds in all the cultures was coumarins (maximum content 1031.5 mg/100 g DW (dry weight), *R. chalepensis*, 5-week growth cycle). The results of the antioxidant tests showed that the extracts of the three species had varied antioxidant capacity: in particular, the *R. chalepensis* extract exhibited the best radical scavenging activity (IC_50_ = 1.665 ± 0.009 mg/mL), while the *R. graveolens* extract displayed the highest chelating property (IC_50_ = 0.671 ± 0.013 mg/mL). Finally, all the extracts showed good activity against *Staphylococcus aureus* with MIC values of 250 μg/mL for the *R. corsica* extract and 500 μg/mL for both *R. graveolens* and *R. chalepensis* extracts.

## 1. Introduction

The genus *Ruta* belongs to the Rutaceae family [1]. The best-known species of the genus are *Ruta graveolens* L. and *Ruta chalepensis* L. Other rue species are less known and may occur, for example, as endemic species such as *Ruta corsica* D. C. [2]. The individual rue species differ slightly in terms of morphology.

*R. graveolens* is a plant whose natural habitats were originally located in south-eastern and eastern Europe, and the species then spread to the area of southern Europe and the African coast of the Mediterranean Sea. *R. chalepensis* is a common species on the Mediterranean coast. It grows on the Italian, Albanian, Greek, Spanish and French coasts, as well as in North Africa (e.g., in Algeria and Libya). It also grows in the areas of India, Latin America, and Western Asia, including countries such as Syria, Iran, Turkey, and Israel. *R. corsica* is a species endemic to the Corsican mountains [3,4].

The chemical composition of the investigated rue species has been studied to varying extents. The chemical composition of *R. graveolens* is well known, while that of *R. corsica* is least known.

The herb of *R. graveolens* is rich in bioactive compounds. It contains different subgroups of coumarins: simple coumarins (coumarin, herniarin, umbelliferon, methoxycoumarin, scopoletin); psoralen-type furanocoumarins (xanthotoxin, bergapten, psoralen, imperatorin, isoimperatorin, isopimpinellin, rutarin), dihydrofuranocoumarins (rutamarine, rutaretin), coumarin dimers (daphnoretin, methoxydaphnoretin), pyranocoumarins (xanthylethine); phenolic acids (vanillic acid, chlorogenic acid, ferulic acid, protocatechuic acid, p-coumaric acid, p-hydroxybenzoic acid, syringic acid, caffeic acid, gentisic acid); alkaloids: acridine (arborinin, rutacridone, gravacridondiol), quinoline (graveolin), furoquinoline (dictamine, cocusaginine, skimmianine, γ-fagarine), dihydrofuroquinoline (ribalin, platydesmin), pyroquinoline (rutaline); flavonoids (rutoside, quercetin, isorhamnetin, kaempferol, myricetin); and essential oil (0.2–0.7%). The simultaneous presence of alkaloids and essential oil is a unique case in the plant kingdom [5,6,7,8].

*R. graveolens* exhibits a broad spectrum of activity, such as antioxidant, anti-inflammatory, spasmolytic, sedative, antibacterial, antifungal, and antidiabetic effects [9,10]. *R. graveolens* is a valuable source of linear furanocoumarins (derivatives of psoralen) that have been used for treating skin diseases such as vitiligo and psoriasis in the so called PUVA therapy (psoralens and UVA irradiation) [11,12].

*R. chalepensis* herb contains secondary metabolites from various chemical groups: coumarins: simple coumarins (coumarin, scopoletin, umbelliferon), coumarin dimers (daphnorin), furanocoumarins (bergapten, chalepensin, chalepin, isopimpinellin, xanthotoxin), dihydrofuranocoumarins (rutamarine, isorutarin); alkaloids: acridine (arborinin, rutacridone), quinoline (graveolin, graveolinin, 3-hydroxygraveoline), furoquinoline (dictamine, isotaifine, cocusaginine, maculosidine, skimmianine, taifine, γ-fagarine, 8-methoxytaifine); flavonoids (hesperidin, rutoside); and essential oil (approximately 0.7%) [13,14,15].

*R. corsica* herb contains isoflavones (daidzein, genistein, prunetin, formononetin). In the root of this species, the following was found: simple coumarins (corsicarin, isoscopoletin), furanocoumarins (psoralen, xanthotoxin, isopimpinellin); furoquinoline alkaloids (skimmianine, dictamine); and essential oil [16,17,18].

In vitro plant cultures can provide high-quality plant material for pharmaceutical or cosmetic purposes. In addition, the use of natural products as screening libraries has recently come of interest and may contribute to the advancement of drug discovery [19]. In vitro cultures of *R. graveolens* are a convenient material for conducting multidirectional research because they exhibit a significant increase in biomass within a short time and show a high capacity for organogenesis and accumulation of many secondary metabolites. Therefore, numerous studies are conducted to determine the quantitative and qualitative composition of the biomass from in vitro cultures of this plant species and subspecies *R. graveolens* ssp. *divaricata* [20,21,22,23,24,25].

The cultures of other *Ruta* species have not been thoroughly investigated. There are few reports on the chemical constituents in *R. chalepensis* in vitro cultures [26,27]. However, there are no studies on the in vitro cultures of *R. corsica* in the available literature. Therefore, the present study aimed to investigate the biosynthetic, antioxidant, and antimicrobial potential of the in vitro cultures of *R. chalepensis* and *R. corsica* and to compare them with the potential of cultures of the most known species, *R. graveolens*.

## 2. Materials and Methods

### 2.1. In Vitro Cultures

The shoot cultures of *Ruta* ssp. were initiated in the Department of Pharmaceutical Botany UJ CM in Cracow: *R. graveolens*—from hypocotyl segments of sterile seedlings derived from the Botanical Garden at the Purkyny University in Brno (Czech Republic); and *R. chalepensis* and *R. corsica*—from seeds derived from the Botanical Garden at the Maria Skłodowska-Curie University in Lublin (Poland).

Stationary liquid cultures were grown on Linsmaier and Skoog medium [28] containing 1 mg/L auxin: NAA (α-naphthaleneacetic acid) and 1 mg/L cytokinin: BAP (6-benzylaminopurine). The experimental agitated cultures were initiated from the stationary liquid cultures.

The cultures were maintained in Erlenmeyer flasks (500 mL), containing 150 mL medium. Initial biomass was 1 g. Cultures were performed on a shaker (Altel) with a rotation frequency of 140/min. The cultures were maintained under artificial light with an intensity of 4 W/m^2^ at 25 ± 2 °C. The cultures were performed in Linsmaier and Skoog (LS) medium with the following content of plant growth regulators: NAA—0.1 mg/L; and BAP—0.1 mg/L. The culturing period was 3, 4, 5, 6, and 7 weeks (3 replicates were performed for each species in each week). After this period the biomass was separated from the medium and dried at 38 °C.

### 2.2. High-Performance Liquid Chromatography Analyses

High-performance liquid chromatography (HPLC) analysis was used to determine the metabolite content of the methanol extracts (sonication, 30 °C, 20 min, three times) obtained from 1 g of dried biomass. RP-HPLC analysis was performed as described elsewhere [29] on a Merck-Hitachi liquid chromatograph (LaChrom Elite, Hitachi, Tokyo, Japan) equipped with a DAD detector L-2455 and a Purospher ^®^ RP-18e 250 × 4 mm/5 μm column (Merck, Darmstadt, Germany). The analysis was conducted at 25 °C, with the mobile phase consisting of A—methanol, B—methanol: 0.5% acetic acid 1:4 (*v*/*v*), gradient elution at the flow rate of 1 mL min^−1^, and the gradient was as follows: 100% B for 0–20 min; 100–80% B for 20–35 min; 80–60% B for 35–55 min; 60–0% B for 55–70 min; 0% B for 70–75 min; 0–100% B for 75–80 min; 100% B for 80–90 min; wavelength range 200–400 nm. The quantification was performed at λ = 254 and 330 nm for phenolic acids, catechins, coumarins, and alkaloids, and at 330 and 370 nm for flavonoids.

The following standards were purchased: bergapten, imperatorin, xanthotoxin, and psoralen from Roth (Karlsruhe, Germany); caffeic acid, chlorogenic acid, cinnamic acid, ellagic acid, gallic acid, gentizic acid, isoferulic acid, neochlorogenic acid, *o*-coumaric acid, protocatechuic acid, rosmarinic acid, salicylic acid, sinapic acid, syringic acid, apigenin, apigetrin (apigenin 7-glucoside), hyperoside (quercetin 3-*O*-galactoside), isoquercetin (quercetin 3-*O*-glucoside), isorhamnetin, kaempferol, luteolin, myricetin, populnin (kaempferol 7-*O*-glucoside), robinin (kaempferol 3-*O*-robinoside-7-*O*-rhamnoside), quercetin, quercitrin (quercetin 3-*O*-rhamnoside), rhamnetin, rutoside, vitexin, 5,7-dimethoxycoumarin, 4-hydroxy-6-methylcoumarin, 6-methylcoumarin, osthole, and umbelliferone from Sigma Aldrich (St Louis, MO, USA); *p*-coumaric acid, vanillic acid, ferulic acid, *p*-hydroxybenzoic acid, coumarin, and scopoletin from Fluka (Bucha, Switzerland), caftaric acid, cryptochlorogenic acid, isochlorogenic acid, catechin, epigallocatechin, epicatechin gallate, epicatechin, epigallocatechin gallate, cinaroside (luteolin 7-*O*-glucoside), osthenol, 4-methylumbelliferone, 4,6-dimethoxy-2H-1-benzopyran-2-one, and skimmianine from ChromaDex (Irvine, CA, USA); 4-*O*-feruloylquinic acid, apigetrin (apigenin 7-*O*-glucoside), apigenin 7-O-glucuronide, astragalin (kaempferol 3-*O*-glucoside), avicularin (quercetin 3-*O*-α-l-arabinofuranoside), trifolin (kaempferol 3-*O*-galactoside), isopimpinellin, isoimperatorin, daphnetin 7-methyl ether, rutaretin, daphnetin, osthenol, bergaptol, daphnetin dimethyl ether, γ-fagarine, and 7-isopentenyloxy-γ-fagarine from ChemFaces (Wuhan, China)

### 2.3. Antioxidant Activity

#### 2.3.1. Free Radical Scavenging Activity

The free radical scavenging activity of the *Ruta* spp. extracts was determined using the DPPH (1,1-diphenyl-2-picrylhydrazyl) method [30]. The extracts were tested at different concentrations (0.0625–2 mg/mL). An aliquot (0.5 mL) of the methanol solution containing different amounts of sample solution was added to 3 mL of daily prepared methanol DPPH solution (0.1 mM). The change in optical density at 517 nm was measured 20 min after the initial mixing by using a model UV-1601 spectrophotometer (Shimadzu). Butylated hydroxytoluene (BHT) was used as a reference.

The scavenging activity was measured as the decrease in the absorbance of the samples versus the DPPH standard solution. Results were expressed as radical scavenging activity percentage (%) of the DPPH, defined by the formula ((A_o_−A_c_)/A_o_) × 100, where A_o_ is the absorbance of the control and A_c_ is the absorbance in the presence of the sample or standard.

The results obtained from the average of three independent experiments are reported as the mean radical scavenging activity percentage (%) ± standard deviation (SD) and mean 50% inhibitory concentration (IC_50_) ± SD.

#### 2.3.2. Reducing Power Assay

The reducing power of the *Ruta* spp. extracts was evaluated by spectrophotometric detection of Fe^3+^-Fe^2+^ transformation [31]. The extracts were tested at different concentrations (0.0625–2 mg/mL). Varying amounts of samples in 1 mL solvent were mixed with 2.5 mL of phosphate buffer (0.2 M, pH 6.6) and 2.5 mL of 1% potassium ferricyanide (K_3_Fe(CN)_6_). The mixture was incubated at 50 °C for 20 min. The resulting solution was cooled rapidly, mixed with 2.5 mL of 10% trichloroacetic acid, and centrifuged at 3000× *g* rpm for 10 min. The resulting supernatant (2.5 mL) was mixed with 2.5 mL of distilled water and 0.5 mL of 0.1% fresh ferric chloride (FeCl_3_), and the absorbance was measured at 700 nm after 10 min. The increased absorbance of the reaction mixture indicates an increase in reducing power. As a blank, an equal volume (1 mL) of water was mixed with a solution prepared as described above. Ascorbic acid and BHT were used as a reference. The results, obtained from the average of three independent experiments, are expressed as the mean absorbance values ± SD. The reducing power was also expressed as the ascorbic acid equivalent (ASE/mL).

#### 2.3.3. Ferrous Ions (Fe^2+^) Chelating Activity

The ferrous (Fe^2+^) ion chelating activity of *Ruta* spp. extracts was estimated by measuring the formation of the Fe^2+^-ferrozine complex, according to the method described previously [32]. The extracts were tested at different concentrations (0.0625–2 mg/mL). Briefly, varying concentrations of each sample in 1 mL solvent were mixed with 0.5 mL of methanol and 0.05 mL of 2 mM FeCl_2_. The reaction was initiated by adding 0.1 mL of 5 mM ferrozine. The mixture was then shaken vigorously and left to stand at room temperature for 10 min. The absorbance of the solution was measured spectrophotometrically at 562 nm. The control contained the FeCl_2_ and ferrozine complex. Ethylenediaminetetraacetic acid (EDTA) was used as a reference. The percentage of inhibition of the ferrozine- (Fe^2+^) complex formation was calculated by the formula ((A_o_ − A_c_)/A_o_) × 100, where A_o_ is the absorbance of the control and A_c_ is the absorbance in the presence of the sample or standard. The results obtained from the average of three independent experiments are reported as mean inhibition of the ferrozine-(Fe^2+^) complex formation (%) ± SD and IC_50_ ± SD.

### 2.4. Antimicrobial Activity

The antimicrobial properties of the *Ruta* spp. extracts were tested against the following microbial strains: *Staphylococcus aureus* ATCC 65381, *Escherichia coli* ATCC 25922, *Pseudomonas aeruginosa* ATCC 9027, and the yeast *Candida albicans* ATCC 10231; these strains were obtained from the in-house culture collection of the Department of Chemical, Biological, Pharmaceutical and Environmental Sciences, University of Messina (Italy). Overnight cultures of the bacterial strains and *C. albicans* were grown at 37 °C in Mueller-Hinton Broth (MHB, Oxoid) and Roswell Park Memorial Institute (RPMI) 1640 medium, respectively. The Minimum Inhibitory Concentration (MIC) and the Minimum Bactericidal/Fungicidal Concentration (MBC/MFC) of the extracts were established according to the guidelines of the Clinical and Laboratory Standards Institute [33,34], with some modifications. The methanol extracts were dissolved in dimethyl sulfoxide (DMSO) and further diluted using MHB or RPMI 1640 to obtain a final concentration of 1 mg/mL. Two-fold serial dilutions were prepared in a 96-well plate. The tested concentrations ranged from 500 to 0.49 μg/mL. Working cultures were adjusted to the required inoculum of 1 × 10^5^ CFU/mL for bacteria and 1 × 10^3^ CFU/mL for yeast. Growth controls (medium with the inoculum but without the extracts) and vehicle controls (medium with the inoculum and DMSO) were also included. The solvent (DMSO) did not exceed 1% concentration. Tetracycline (TC), as a positive control, was tested against all the bacterial strains at concentrations ranging from 32 to 0.016 μg/mL, while itraconazole (ITC) was tested against the yeast. The MIC of TC was considered as the lowest concentration of the extracts at which there was no bacterial growth. The MIC of ITC was defined as the lowest drug concentration that inhibited ≥50% growth as compared to the control [35]. To determine MBC/MFC values, bacterial or yeast aliquots (10 µL) were taken from each well and cultured on Mueller-Hinton Agar (MHA, Oxoid) or Sabouraud Dextrose Agar (SDA, Oxoid), respectively. The cultures were incubated for 24–48 h at 37 °C.

### 2.5. Statistical Analysis

Statistical comparison of data was performed using one-way analysis of variance (ANOVA) followed by Tukey–Kramer multiple comparisons test (GraphPAD Prism Software for Science LLC, San Diego, CA, USA). *p*-values less than 0.05 were considered statistically significant.

## 3. Results and Discussion

### 3.1. Biomass Increments

The in vitro cultures of *R. graveolens*, *R. corsica*, and *R. chalepensis* were maintained as agitated shoot cultures in a 7-week growth cycle (inoculum mass: 1 g). The cultures were characterized by a similar increase in biomass. The cultures showed a good biomass growth up to the fifth week of cultivation, except for *R. graveolens* cultures, which reached their maximum growth after four weeks of breeding. In the sixth week the growth gradually stopped, and after 7 weeks of cultivation, the shoots started to show signs of dieback. The best (33.7-fold) increase in biomass was observed in *R. chalepensis* cultures after 5 weeks of cultivation. For *R. graveolens* and *R. corsica*, the following biomass increase was recorded: 33.3-fold (after 4 weeks) and 31.3-fold (after 5 weeks), respectively (Figure 1).

### 3.2. Accumulation of Bioactive Metabolites

The HPLC analysis of methanolic extracts obtained from the microshoots of the investigated in vitro cultures of the three rue species confirmed the accumulation of linear furanocoumarins (bergapten retention time (RT) = 63.72 min, isoimperatorin RT = 70.55 min, isopimpinellin RT = 60.91 min, psoralen RT = 53.38 min, xanthotoxin RT = 54.18 min) and furoquinoline alkaloids (γ-fagarine RT = 64.68 min, 7-isopentenyloxy-γ-fagarine RT = 71.92 min, skimmianine RT = 62.36 min). Sample chromatograms of the extracts from the *R. corsica*, *R chalepensis*, and *R. graveolens* in vitro cultures are included in Appendix A.

No flavonoids were detected. Phenolic acids were accumulated in very small amounts. The dynamics of accumulation of individual metabolites varied according to the time of the culture and species. Generally, the production of most compounds increased between the third and sixth week and decreased in the seventh week. Table 1 shows the mean contents (mg/100 g DW) of individual metabolites depending on the rue species and the growth cycle. Among the coumarins, xanthotoxin was accumulated in the greatest amount (maximum content: *R. chalepensis*—509.8 mg/100 g DW; *R. corsica*—375.9 mg/100 g DW, 5-week growth cycle; *R. graveolens*—428.3 mg/100 g DW, 6-week growth cycle). Bergapten was the next coumarin that was accumulated in high amounts with the maximum content obtained after the 6-week growth cycle in *R. chalepensis* and *R. graveolens* cultures and after the 7-week growth cycle in *R. corsica* cultures (281.4; 186.6; 174.7 mg/100 g DW, respectively). The quantity of psoralen was also high, with the maximum content of 268.8 mg/100 g DW in *R. chalepensis* microshoots (5-week growth cycle) and 278.7 and 129.3 mg/100 g DW in *R. corsica* and *R. graveolens* microshoots (6-week growth cycle). The amounts of isopimpinellin were lower and reached the maximum content of 145.25 and 57.45 mg/100 g DW in *R. graveolens* and *R. corsica* extracts of biomass (5-week growth cycle), respectively, and 77.1 mg/100 g DW in the *R. chalepensis* extract (6-week growth cycle). Isoimperatorin was accumulated in the lowest amount of 46.8, 43.1, and 23.9 mg/100 g DW in *R. corsica*, *R. chalepensis*, and *R. graveolens* microshoots, respectively (6-week growth cycle).

Among the confirmed furoquinoline alkaloids, γ-fagarine was accumulated in the highest amount (293.7 mg/100 g DW in *R. corsica* cultures after the 4-week growth cycle, while its content was 155.9 and 124.3 mg/100 g DW in *R. graveolens* and *R. chalepensis* cultures after the 5-week growth cycle, respectively). The maximum skimmianine content was 133.4 mg/100 g DW in *R. corsica* cultures (4-week growth cycle) 94.6 mg/100 g DW in *R. graveolens* cultures (5-week growth cycle) and 55.5 mg/100 g DW in *R. chalepensis* cultures (7-week growth cycle). Isopentenyloxy-γ-fagarine was accumulated in the lowest amount: 26.5 mg/100 g DW in *R. corsica* microshoots (4-week growth cycle) and 6.8 and 4.4 mg/100 g DW in *R. graveolens* and *R. chalepensis* microshoots, respectively (5-week growth cycle) (Table 1).

The highest total content (1031.5 mg/100 g DW) of the estimated coumarins was detected in the cultures of *R. chalepensis* (5-week growth cycle). Very high amounts of coumarins were also confirmed in the cultures of *R. graveolens* (917.23 mg/100 g DW, 5-week growth cycle) and *R. corsica* (878.05 mg/100 g DW, 6-week growth cycle) (Table 1). During the breeding cycle, the total content of coumarins in the biomass of *R. chalepensis* cultures increased from 617.5 mg/100 g DW (3-week growth cycle) to 1031.5 mg/100 g DW (5-week growth cycle) (1.38-fold), in the biomass of *R. graveolens* cultures from 461.4 mg/100 g DW (3-week growth cycle) to 917.2 mg/100 g DW (5-week growth cycle) (1.99-fold), and in the biomass of *R. corsica* cultures from 533.5 mg/100 g DW (3-week growth cycle) to 878.05 mg/100 g DW (6-week growth cycle) (1.65-fold) (Table 1, Figure 2).

Shoot cultures of *R. corsica* showed the highest ability to accumulate alkaloids. The highest total content (293.7 mg/100 g DW) of the estimated alkaloids was detected after the 4-week growth cycle and was 1.88-fold higher than the maximal content of alkaloids in *R. graveolens* cultures (155.9 mg/100 g DW, 5-week growth cycle) and 2.36-fold higher than the maximal content of alkaloids in *R. chalepensis* cultures (124.3 mg/100 g DW, 5-week growth cycle) (Table 1). During the breeding cycle the total content of alkaloids in the biomass of *R. corsica* cultures increased from 175.4 mg/100 g DW (3-week growth cycle) to 293.7 mg/100 g DW (5-week growth cycle) (1.67-fold), in the biomass of *R. graveolens* cultures from 105.1 mg/100 g DW (3-week growth cycle) to 155.9 mg/100 g DW (5-week growth cycle) (1.48-fold), and in the biomass of *R. chalepensis* cultures from 61.25 mg/100 g DW (3-week growth cycle) to 124.3 mg/100 g DW (5-week growth cycle) (2.03-fold) (Table 1, Figure 3).

The production of secondary metabolites from the group of linear furanocoumarins and some alkaloids was observed in the in vitro cultures of the studied rue species. A characteristic of in vitro cultures is that their metabolic profile may differ from that of their parent plants. Our previous studies on in vitro cultures of *R. graveolens* confirmed that the shoot cultures of this plant species accumulated mainly linear furanocoumarins and alkaloids, and significantly lower amounts of phenolic acids [21,23]. The results of current studies also confirmed this ability of the *R. graveolens* shoot culture. The same biosynthetic potential was now documented by us for two other rue species—*R. chalepensis* and *R. corsica*. The shoots of these species also produced mainly linear furanocoumarins and some alkaloids. However, some differences were noted between the in vitro cultures of the three rue species. *R. corsica* shoots produced higher amounts of alkaloids than *R. chalepensis* and *R. graveolens* shoot cultures.

Previous studies [36] analysed the selected secondary metabolites (coumarins and alkaloids) in *R. graveolens* shoot stationary liquid cultures (LS medium containing 2 mg/L each of NAA and BAP). Seven secondary metabolites were identified. These included linear furanocoumarins (psoralen, bergapten, xanthotoxin, isopimpinellin), linear dihydrofuranocoumarin (rutamarine), and furoquinoline alkaloids (cocusaginine and skimmianine). Five of these were confirmed in the current HPLC analysis of *R. graveolens* biomass extracts. Additionally, we confirmed two furoquinoline alkaloids (γ-fagarine, isopentenyloxy-γ-fagarine) and isoimperatorin (furanocoumarin).

The results of dynamics of metabolites accumulation during the growth cycles showed that the maximal total amounts of furanocoumarins were obtained after the 5-week growth cycle. In our previous studies on agitated shoot cultures of *R. graveolens*, which were grown in a 6-week growth cycle, the maximal content (520.8 mg/g DW) of linear furanocoumarin was confirmed after 6 weeks (LS medium NAA/BAP 0.1/0.1 mg/L) [21].

The maximal accumulation (966 mg/100 g DW) of furanocoumarins in the stationary liquid culture of *R. graveolens* was confirmed after 4 weeks of growth (LS medium NAA/BAP: 2/2 mg/L) [20].

Baumert et al. showed the presence of the following metabolites in in vitro callus agar culture of *R. chalepensis*: furanocoumarins: bergapten and isopimpinellin; alkaloids: skimmianine, γ-fagarine, rutacridone, rutacridone epoxide, gravacridone, and arborinin. The authors, however, did not quantify these compounds [26].

In the study of Fischer et al., the authors isolated and identified three metabolites from *R. chalepensis* suspension cultures: isorutarin (dihydrofuranocoumarin), rutarensin—an unknown ester of daphnorin (coumarin dimer) and 3-hydroxy-3-methylglutaric acid. The authors, however, did not isolate the linear furanocoumarins detected in our present study [27].

As part of this work, the *R. chalepensis* shoot cultures were conducted for the first time. The abovementioned authors investigated other types of cultures (callus and suspension cultures). A comparison of the results of previous studies with those obtained in the present study revealed significant qualitative differences in the accumulated compounds from the coumarin group between different types of cultures. The shoot cultures studied in the present research accumulated high amounts of xanthotoxin and psoralen. However, they were not found in callus and suspension cultures.

There are no studies on in vitro cultures of the endemic species *R. corsica* in the available literature. In the present study, the metabolic profile of the *R. corsica* shoot cultures was investigated for the first time.

On the basis of the obtained results, all the studied rue cultures can be proposed as an alternative, rich, in vitro-controlled source of linear furanocoumarins. Xanthotoxin and bergapten in particular are objects of interest in dermatology. These compounds have photosensitizing effects on human skin and are used for their pigmentation stimulating and antiproliferative properties in the symptomatic treatment of vitiligo, psoriasis, and mycosis fungoides [37]. The obtained results showed very high levels of xanthotoxin in *R. chalepensis* and *R. graveolens* cultures (above 500 and 420 mg/100 g DW) and approximately 400 mg/100 g DW in *R. corsica* cultures. The obtained maximal amounts of bergapten were also high (above 280 mg/100 g DW in *R. chalepensis* cultures, above 180 mg/100 g DW in *R. graveolens* cultures, and above 170 mg/100 g DW in *R. corsica* cultures). For comparison, the content of xanthotoxin and bergapten in plants cultivated in the field is as follows: for *R. graveolens*—xanthotoxin 100 mg/100 g DW, bergapten 160 mg/100 g DW; for *R. chalepensis*—xanthotoxin 10 mg/100 g DW, bergapten 260 mg/100 g DW [29]. The biosynthetic potential for the production of furoquinoline alkaloids is the highest for *R. corsica* cultures (above 130 mg/100 g DW). These cultures can therefore be considered as a source of furoquinoline alkaloids: γ-fagarine and skimmianine. Furoquinoline alkaloids show a number of biological activities, including antifungal and antibacterial properties, inhibitory activity towards AchE (acetylcholinesterase), and 5-HT2 receptor-inhibiting properties [38].

### 3.3. Antioxidant Activity

Antioxidants compounds neutralize free radicals and their negative effects by preventing their production, intercepting them, and repairing the damage caused by these molecules; these compounds use various mechanisms, such as functioning as reducing agents by trapping free radicals, donating hydrogen, acting as chelators, and quenching singlet oxygen. Antioxidants can be divided into primary (or chain-breaking) and secondary (or preventive), and the primary antioxidant reactions can be classified as hydrogen-atom transfer (HAT) and single-electron transfer (SET). The HAT mechanism occurs when an antioxidant scavenges free radicals by donating hydrogen atoms; while in the SET mechanism, an antioxidant acts by transferring a single electron to reduce any compound.

Antioxidants can act through various mechanisms, such as binding of transition metal ion catalysts, reducing capacity and radical scavenging, prevention of chain initiation, decomposition of peroxides, and prevention of continued hydrogen abstraction. Therefore, it is important to use methodologies with distinct mechanisms to evaluate the antioxidant capacity of plant-derived phytocomplexes or isolated compounds [39]. 

To determine the in vitro antioxidant capacity of methanolic extracts of *Ruta* spp., three in vitro tests based on different mechanisms of determination of the antioxidant capacity were used: the DPPH (1,1-diphenyl-2-picrylhydrazyl) method, (involving both HAT and SET mechanisms); the reducing power assay (based on SET mechanism); and the ferrous ion chelating activity assay, to assess secondary antioxidant ability. 

The results of the DPPH assay showed that all *Ruta* spp. extracts exhibited the radical scavenging effect that increased with the increasing concentrations (Figure 4). The *R. chalepensis* extract showed the best radical scavenging properties, reaching 60% of the activity at the higher concentration tested (Figure 4). As indicated by the IC_50_ values, the activity of the extracts was found to be lower than that of butylated hydroxytoluene (BHT) used as a standard drug and it decreased in the following order: BHT > *R. chalepensis > R. graveolens > R. corsica* (Table 2).

The antiradical activity of the extracts obtained from *R. chalepensis* grown in different countries has been extensively studied. Fakhfakh et al. (2012) found IC_50_ values of 0.12 and 0.22 mg/mL for *R. chalepensis* ethanol and aqueous extracts, respectively [40]. Ereifej et al. (2015) reported a DPPH radical scavenging ability of Tunisian leaf extract with an IC_50_ value of 70.01 μg/mL [41]. Similar results were obtained by Rached et al. (2010) who reported an IC_50_ value of 61.41 μg/mL for an Algerian *R. chalepensis* leaf ethanol extract and by Loizzo et al. (2018) who reported an IC_50_ value of 60.2 μg/mL for the methanol leaf extract obtained from Tunisian samples of this species [42,43]. Kacem et al. (2015) highlighted a decreasing DPPH radical scavenging potential in different extracts of *R. chalepensis* from Tunisia: ethanol extract > methanol/water (1/1) extract > methanol extract > ethyl acetate extract > water extract > hexane extract [14]. In contrast, other authors found a higher IC_50_ value for the methanolic extract of the aerial part of *R. chalepensis* from Saudi Arabia (320.7 μg/mL) [44]. Interestingly, the IC_50_ values in the DPPH test were found for decoction and ethanol extracts of stem and leaves of *R. chalepensis* from El Hamma (Tunisia) (IC_50_ value of 2.26 μg/mL for the leaf decoction) [45]. 

Regarding the antioxidant activity of *R. graveolens*, Molnar et al. (2017) reported that the ethanolic extracts obtained from commercial samples of the plant showed a higher DPPH scavenging activity (% inhibition = 59.3%) than the hexane extract (% inhibition = 16.8%) at the same concentration (250 μg/mL) [46]. Furthermore, methanolic and ethanolic extracts of wild-growing and cultivated Serbian *R. graveolens* at the beginning and end of the flowering season exhibited significant antioxidant potential. The methanolic extract of wild plants collected at the end of the flowering season showed the strongest antioxidant activity in DPPH tests (IC_50_ = 36.36 μg/mL) [47].

As shown in Figure 5, the *Ruta* spp. extracts showed a weak, dose-dependent reducing power. A comparison of the ASE/mL values showed that the reducing power of the *Ruta* spp. extracts and the standard decreased in the following order: BHT > *R. chalepensis > R. corsica > R. graveolens* (Table 2).

In the Fe^2+^ chelating activity assay, the *Ruta* spp. extracts exhibited good, dose-dependent chelating properties. *R. graveolens* showed the best activity, reaching approximately 80% at the highest tested concentration (Figure 6). Based on the IC_50_ values, the activity of the *Ruta* spp. extracts and the standard decreased in the following order EDTA > *R. graveolens > R. corsica > R. chalepensis* (Table 2). Loizzo et al. (2018) conducted a study on the methanolic leaf extract of Tunisian *R. graveolens* and found that this extract showed low metal chelating activity (14.6%) as compared to EDTA (97.8%) [43].

On the basis of the obtained results, it is evident that the methanolic extracts of the *Ruta* spp. cultures act as weak primary antioxidants and possess good secondary antioxidant properties.

The genus *Ruta* is rich in bioactive compounds. Phytochemical investigations conducted on extracts obtained from *Ruta* spp. grown in fields have detected the presence of compounds belonging to different chemical classes, such as coumarins, phenolic acids, flavonoids, alkaloids, and tannins [48].

In our HPLC study of in vitro cultures of *Ruta* spp., a different metabolic profile with respect to that of the parent plant was highlighted. In all the extracts, flavonoids were undetectable and phenolic acids were present only in small amounts. The dominant group of compounds in all cultures were coumarins followed by alkaloids. Among the coumarins, xanthotoxin was accumulated in the highest amount. It was reported that coumarins imperatorin, xanthotoxin, and bergapten did not exhibit anti-radical activity against DPPH, while they showed a moderate level of reducing power, moreover, they were effective in the ferrous ion-chelation test [49]. Coumarins can reduce the level of oxidative stress through chelation of redox-active Cu and Fe, thus suppressing the formation of ROS through the Fenton reaction [50].

Thus, the good secondary antioxidant properties observed for *R. graveolens*, *R. Corsica*, and *R. chalepensis* methanol extracts could be mainly related to the presence of coumarins, but it cannot be excluded that other compounds present in the phytocomplex could also contribute to the observed effects.

### 3.4. Antimicrobial Activity

The development of antimicrobial resistance in pathogens has prompted extensive research to find alternative therapeutics. *Staphylococcus aureus* is the first pathogen that has become resistant to all known antibiotics and thus it is essential that new antimicrobial agents are discovered to combat this problem [51]. Plants rich in secondary metabolites are one of the go-to reservoirs for the discovery of potential resources to alleviate this problem [52,53]. 

The preliminary antimicrobial screening of the *Ruta* spp. extracts obtained from in vitro cultured biomass was performed against a small representative set of Gram-positive and Gram-negative bacterial strains and the yeast *Candida albicans* according to the protocols recommended by the Clinical and Laboratory Standards Institute [33,34]. The obtained results showed that all the methanolic extracts had a good bacteriostatic activity against the Gram-positive *S. aureus* at the tested concentrations. Among the extracts, the *R. corsica* extract showed the highest activity (MIC = 250 μg/mL) followed by *R. graveolens* and *R. chalepensis* extracts (MIC = 500 μg/mL). None of the tested strains were inhibited by DMSO (maximum 0.5% *v*/*v*), used as a negative control. The MIC of tetracycline, used as positive control, was 1 μg/mL.

Several studies have reported the antibacterial and antifungal potential of *Ruta* spp., with *R. chalepensis* and *R. graveolens* being the most frequently investigated [48]. 

Our results showed good antibacterial activity of the methanolic extract obtained from *Ruta* spp. in vitro cultured biomass against *S. aureus* at concentrations lower (MIC range: 250–500 μg/mL) than those reported by Alzoreky and Nakahara (2003) for wild or cultivated *R. chalepensis* and *R. graveolens* (MIC = 2640 mg/mL) [54]. There are no data on *R. corsica*.

The in vitro cultures studied in the present research accumulated mainly metabolites from the group of coumarins and alkaloids. Alkaloids exhibit antimicrobial activity by enzyme activity inhibition, intracellular molecular imbalance, and cell membrane disturbances, which can cause cell death [55]. All the extracts obtained from *Ruta* spp. in vitro cultured biomass contained furoquinoline alkaloids, namely, γ-fagarine, 7-isopentenyloxy-γ-fagarine, and skimmianine, whose strong antibacterial activity against *S. aureus* has been widely demonstrated [56,57,58]. 

The highest total content of the identified alkaloids was found in the *R. corsica* extract, which verified the greater activity shown by this extract than the other two extracts. 

Coumarins possess a wide variety of biological activities [59]; however, only a limited number of coumarin structures have been assessed for antimicrobial activity [48]. Among these, the furanocoumarin xanthotoxin proved to be effective in inhibiting the growth of *S. aureus* while bergapten was not active [53]. 

Thus, it can be hypothesized that the good antibacterial activity observed for all *Ruta* spp. extracts against *S. aureus* could depend mainly on the presence of the alkaloids γ-fagarine and skimmianine and the coumarin xanthotoxin contained in the phytocomplex.

## 4. Conclusions

The present research proved a similar biosynthetic potential of shoot cultures of three rue species: *R. chalepensis*, *R. corsica*, and *R. graveolens*. The metabolism of the in vitro cultures of these species differs from that of the parent plants and is mainly directed towards the production of linear furanocoumarin compounds and furoquinoline alkaloids. The antioxidant and antibacterial activities of all rue cultures were confirmed. These activities are derived from the presence of the dominant metabolites from the furanocoumarin group and alkaloids, but they may also be related to the action of other compounds present in the phytocomplex. The obtained contents of the described groups of metabolites are higher than those in plants grown in the field. Therefore, in vitro cultures of *Ruta* spp. can be proposed as an alternative source of obtaining these metabolites: all rue cultures tested can be used to obtain furanocoumarins, while the most effective source for obtaining alkaloids is an *R. corsica* culture. 

## Figures and Tables

**Figure 1 antioxidants-11-00592-f001:**
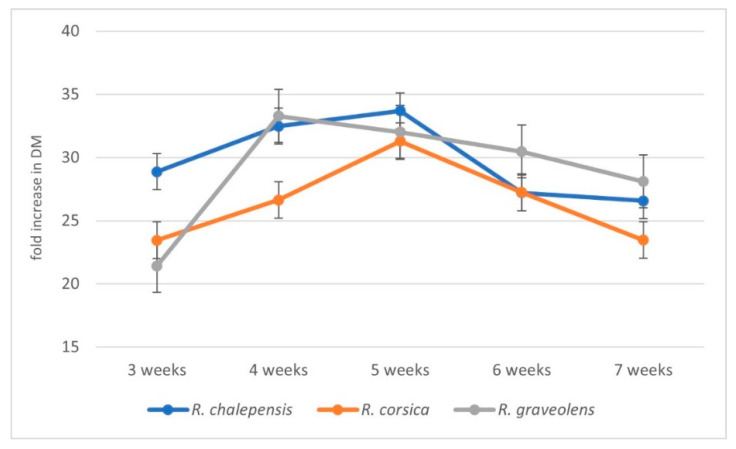
Average (*n* = 3, ±SD) biomass increments during the cultivation cycle (3, 4, 5, 6, and 7 weeks) of individual rue species.

**Figure 2 antioxidants-11-00592-f002:**
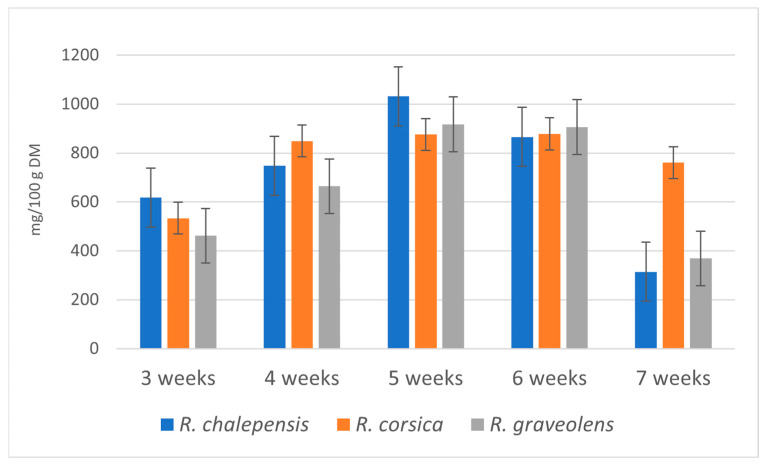
Total content of coumarins (mg/100 g DW) in the methanol extracts obtained from *Ruta* spp. in vitro cultured biomass in the consecutive weeks of the breeding cycle. Values are expressed as the mean ± SD (*n* = 3).

**Figure 3 antioxidants-11-00592-f003:**
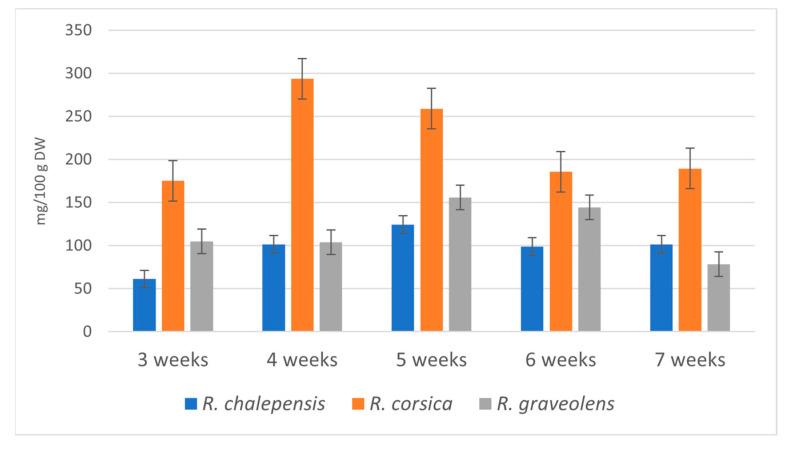
Total content of alkaloids (mg/100 g DW) in the methanol extracts obtained from the *Ruta* spp. in vitro cultured biomass in the consecutive weeks of the breeding cycle. Values are expressed as the mean ± SD (*n* = 3).

**Figure 4 antioxidants-11-00592-f004:**
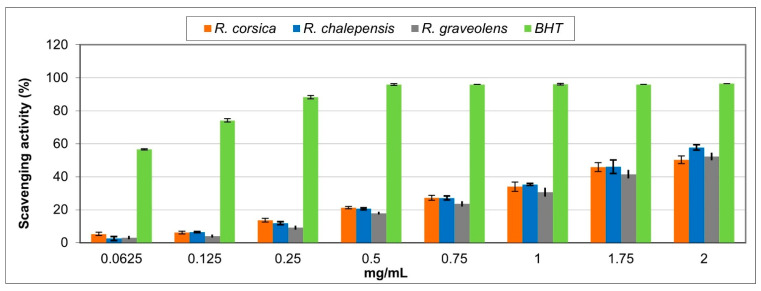
Free radical scavenging activity of the methanolic extracts of the *Ruta* spp. in vitro cultured biomass. Values are expressed as the mean ± SD (*n* = 3).

**Figure 5 antioxidants-11-00592-f005:**
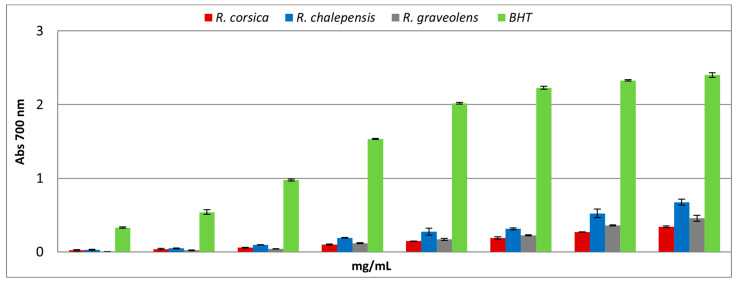
Reducing power of the methanolic extracts of the *Ruta* spp. in vitro cultured biomass, evaluated by spectrophotometric detection of Fe^3+^-Fe^2+^ transformation. Values are expressed as the mean ± SD (*n* = 3).

**Figure 6 antioxidants-11-00592-f006:**
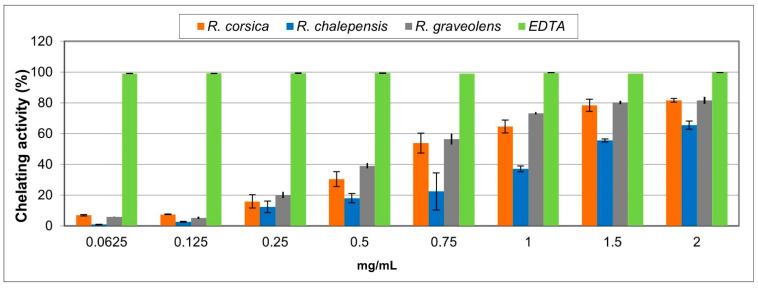
The chelating activity of the methanolic extracts of the *Ruta* spp. in vitro cultured biomass measured by inhibition of the ferrozine-Fe^2+^ complex formation. Values are expressed as the mean ± SD (*n* = 3).

**Table 1 antioxidants-11-00592-t001:** Average contents of metabolites (mg/100 g DW) in methanol extracts obtained from biomass of cultures of three rue species depending on the duration of the culture (3, 4, 5, 6, and 7 weeks). Means of three measurements ± SD. Different letters indicate significant differences (*p* < 0.05).

Accumulated Compounds	Growth Period	*R. chalepensis*	*R. corsica*	*R. graveolens*
**bergapten**	3 weeks	91.537 ± 11.656 ^ace^	89.049 ± 13.492 ^ace^	55.391 ± 3.004 ^be^
	4 weeks	112.803 ± 8.821 ^adl^	135.548 ± 10.760 ^cdghl^	78.183 ± 6.148 ^abe^
	5 weeks	184.083 ± 10.970 ^fhjk^	149.363 ± 6.694 ^dghjl^	162.236 ± 9.359 ^dfghj^
	6 weeks	**281.388 ± 8.330 ^i^**	165.047 ± 7.075 ^fghk^	186.603 ± 2.905 ^fhjk^
	7 weeks	130.150 ± 2.211 ^cdgl^	174.666 ± 12.736 ^fghjk^	67.990 ± 12.442 ^abe^
**isoimperatorin**	3 weeks	15.128 ± 1.156 ^abh^	23.696 ± 4.974 ^abcd^	14.611 ± 1.951 ^abdh^
	4 weeks	20.944 ± 2.184 ^abcd^	32.090 ± 10.191 ^bcdef^	23.299 ± 1.807 ^abcd^
	5 weeks	28.573 ± 3.600 ^bcde^	36.684 ± 2.145 ^cdefg^	22.003 ± 4.661 ^abcd^
	6 weeks	43.126 ± 1.183 ^cefg^	**46.771 ± 1.435 ^efg^**	23.931 ± 1.068 ^abcd^
	7 weeks	22.742 ± 2.426 ^abcd^	42.981 ± 4.047 ^cefg^	6.610 ± 1.690 ^ah^
**isopimpinellin**	3 weeks	37.867 ± 9.467 ^ab^	43.184 ± 1.851 ^abd^	92.809 ± 1.865 ^c^
	4 weeks	31.680 ± 1.116 ^ab^	47.130 ± 1.859 ^abd^	76.459 ± 2.762 ^c^
	5 weeks	40.157 ± 2.292 ^ab^	57.455 ± 1.945 ^bd^	**145.251 ± 7.313 ^de^**
	6 weeks	77.140 ± 9.462 ^c^	46.760 ± 1.668 ^abd^	138.345 ± 4.211 ^e^
	7 weeks	36.063 ± 5.455 ^ab^	44.033 ± 10.052 ^abd^	84.774 ± 8.748 ^c^
**psoralen**	3 weeks	122.970 ± 15.264 ^abcdg^	98.344 ± 0.902 ^abcgi^	70.121 ± 9.557 ^abcgih^
	4 weeks	177.943 ± 27.752 ^ade^	246.241 ± 51.342 ^def^	100.141 ± 9.547 ^abcgi^
	5 weeks	268.823 ± 6.085 ^ef^	256.226 ± 37.091 ^ef^	105.269 ± 5.043 ^abcg^
	6 weeks	98.766 ± 3.088 ^abcgi^	**278.699 ± 26.560 ^ef^**	129.266 ± 3.940 ^abcdg^
	7 weeks	24.047 ± 1.514 ^chi^	179.779 ± 23.994 ^ade^	37.421 ± 22.530 ^bchi^
**xanthotoxin**	3 weeks	349.997 ± 54.203 ^abdehijk^	279.242 ± 44.171 ^abcj^	228.435 ± 11.563 ^bcjk^
	4 weeks	404.187 ± 42.136 ^adeghj^	397.999 ± 19.047 ^adehj^	385.553 ± 24.113 ^adehj^
	5 weeks	**509.8** **27 ± 14.507 ^fgh^**	375.924 ± 54.065 ^adehj^	385.480 ± 18.961 ^dfgh^
	6 weeks	365.223 ± 20.883 ^abdehj^	340.774 ± 10.539 ^abdehj^	428.302 ± 19.060 ^adefgh^
	7 weeks	101.527 ± 2.700 ^ik^	319.537 ± 27.264 ^abcdej^	171.908 ± 45.753 ^cik^
**Total coumarins**	3 weeks	617.498 ± 28.808 ^abf^	533.515 ± 32.612 ^abc^	461.368 ± 9.025 ^bcm^
	4 weeks	747.557 ± 59.484 ^defl^	849.022 ± 85.331 ^dehij^	663.636 ± 32.960 ^adfl^
	5 weeks	**1031.463 ± 21.196 ^gi^**	875.651 ± 40.408 ^ehijl^	917.230 ± 32.550 ^eghijl^
	6 weeks	865.643 ± 26.639 ^ehijl^	878.051 ± 40.412 ^ehij^	906.447 ± 15.819 ^ehij^
	7 weeks	314.530 ± 5.806 ^kl^	760.995 ± 40.617 ^defhl^	368.703 ± 26.781 ^ckm^
**γ-fagarine**	3 weeks	41.203 ± 9.563 ^acfhil^	83.388 ± 5.113 ^bdjk^	47.164 ± 5.480 ^acdfhil^
	4 weeks	65.350 ± 5.152 ^bcdhik^	**133.804 ± 7.766 ^eg^**	34.663 ± 0.728 ^acfhl^
	5 weeks	78.810 ± 6.455 ^bdjk^	114.682 ± 10.950 ^egj^	54.517 ± 3.776 ^acdhi^
	6 weeks	58.341 ± 2.812 ^acdhik^	97.303 ± 6.759 ^bgj^	53.649 ± 0.803 ^acdfhi^
	7 weeks	42.638 ± 5.295 ^acfhil^	74.105 ± 10.043 ^bdik^	31.106 ± 4.124 ^acfl^
**Isopentenyloxy-γ-fagarine**	3 weeks	2.778 ± 0.714 ^aefh^	17.482 ± 1.638 ^b^	9.143 ± 0.503 ^ceg^
	4 weeks	2.935 ± 0.100 ^aefh^	**26.505 ± 2.900 ^d^**	6.311 ± 0.705 ^acefg^
	5 weeks	4.379 ± 0.357 ^aefgh^	20.397 ± 1.339 ^b^	6.788 ± 0.027 ^cefg^
	6 weeks	3.015 ± 1.010 ^aefh^	25.669 ± 0.066 ^d^	5.479 ± 0.100 ^acefg^
	7 weeks	3.337 ± 0.498 ^aefgh^	19.931 ± 2.741 ^b^	1.644 ± 0.088 ^afh^
**skimmianine**	3 weeks	17.267 ± 5.937 ^ad^	74.520 ± 11.342 ^bfi^	48.836 ± 2.680 ^cdfgj^
	4 weeks	33.336 ± 2.404 ^acdg^	**133.412 ± 1.278 ^e^**	63.099 ± 5.734 ^bcfj^
	5 weeks	41.127 ± 6.753 ^cdgj^	123.816 ± 10.221 ^e^	94.635 ± 5.419 ^hi^
	6 weeks	37.608 ± 4.668 ^cdgj^	62.912 ± 1.729 ^bcfj^	85.399 ± 3.297 ^bhi^
	7 weeks	55.535 ± 5.968 ^cfgj^	95.575 ± 9.810 ^hi^	45.517 ± 3.340 ^cdfgj^
**Total alkaloids**	3 weeks	61.247 ± 13.462 ^aj^	175.390 ± 9.103 ^bgh^	105.143 ± 6.920 ^cej^
	4 weeks	101.621 ± 4.374 ^cej^	**293.721 ± 8.627 ^d^**	104.074 ± 6.286 ^cej^
	5 weeks	124.316 ± 12.860 ^cei^	258.895 ± 12.770 ^f^	155.940 ± 7.130 ^bgi^
	6 weeks	98.964 ± 6.569 ^cej^	185.883 ± 5.235 ^bh^	144.527 ± 2.506 ^egi^
	7 weeks	101.510 ± 5.749 ^cej^	189.612 ± 20.186 ^bh^	78.267 ± 3.789 ^acj^

**Table 2 antioxidants-11-00592-t002:** Free radical scavenging activity (DPPH test), reducing power, and ferrous ions (Fe^2+^) chelating activity of the methanolic extracts of the *Ruta* spp. in vitro cultured biomass.

Extracts	DPPH TestIC_50_ (mg/mL)	Reducing Power AssayASE/mL	Fe^2+^ Chelating ActivityIC_50_ (mg/mL)
*Ruta chalepensis*	1.665 ± 0.009 ^a^	15.493 ± 0.207 ^a^	1.452 ± 0.012 ^a^
*Ruta corsica*	2.032 ± 0.001 ^b^	20.516 ± 0.016 ^b^	0.736 ± 0.100 ^b^
*Ruta graveolens*	1.883 ± 0.007 ^c^	26.012 ± 0.019 ^c^	0.671 ± 0.013 ^a c^
Standard	BHT0.065 ± 0.008 ^d^	BHT1.131 ± 0.037 ^d^	EDTA0.0067 ± 0.0003 ^d^

Values are expressed as the mean ± SD. (*n* = 3). ^a–d^ Different letters within the same column indicate significant differences between mean values (*p* < 0.05).

## Data Availability

The data presented in this study are available in the article and Appendix A.

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
