# Peer review of "Phytochemical Characterization, and Antioxidant and Antimicrobial Properties of Agitated Cultures of Three Rue Species: Ruta chalepensis, Ruta corsica, and Ruta graveolens"

_antioxidants, 2022, doi:10.3390/antiox11030592_

Round 1

Reviewer 1 Report

This manuscript written by Szewczyk Agnieszka et al. shows the phytochemical characterization of methanol extracts obtained from biomass of in vitro cultures of three rue species, Ruta chalepensis, R. corsica, and R. graveolens, and their antioxidant capacity. In addition, the authors show the data on a preliminary screening of their antimicrobial potential.

Overall, the manuscript is well written, and the research topic is novel. This reviewer has a couple of minor comments on this manuscript.

Minor comments

Dimethyl sulfoxide is wrongly abbreviated.

Line 190-192: Did the authors dissolve the methanol extracts in 1% DMSO? If so, is 99% water? Not in absolute DMSO?

The authors need to show the gradient condition of HPLC analysis. In addition, retention time of each standard should be shown.

Author Response

Dear Reviewer,

We are greatly obliged for having received the Reviewers’ valuable opinion and helpful suggestion on our manuscript. All changes in the manuscript are marked in yellow and blue. The replies to the specific comments are listed below.

Dimethyl sulfoxide is wrongly abbreviated.

Answer: We correct the abbreviation DMSO.

Line 190-192: Did the authors dissolve the methanol extracts in 1% DMSO? If so, is 99% water? Not in absolute DMSO?

Answer: Yes, the methanol extracts were dissolved in dimethyl sulfoxide (DMSO) and further diluted using MHB or RPMI 1640 culture media to obtain a final concentration of 1 mg/mL. The solvent (DMSO) did not exceed 1%, because for microbial cells more than 1% of DMSO can be toxic.

In the manuscript the sentence has been correct as follow: “The methanol extracts were dissolved in DMSO and further diluted using MHB or RPMI 1640 to obtain a final concentration of 1 mg/mL. Two-fold serial dilutions were prepared in a 96-well plate. The tested concentrations ranged from 500 to 0.49 μg/mL. Working cultures were adjusted to required inoculum of 1 × 105 CFU/mL for bacteria and 1 × 103 CFU/mL for yeast. Growth controls (medium with inoculum but without extracts) and vehicle controls (medium with inoculum and DMSO) were also included. The solvent (DMSO) did not exceed 1% concentration.

The authors need to show the gradient condition of HPLC analysis.

Answer: The gradient condition has been added.

In addition, retention time of each standard should be shown.

Answer: Retention times of standards have been added.

Please, accept my best regards,

Yours sincerely,

Agnieszka Szewczyk

Reviewer 2 Report

Comments: Obviously, this manuscript reported by Agnieszka et al is clear to understand. In this work, active compounds from the Ruta: R. chalepensis, R. corsica R. graveolens were identified successfully. The antioxidant and antimicrobial activities were also investigated. In vitro experimental testing demonstrated that several active ingredients were exhibited radical scavenging activity. Minor revisions noted to the authors and then I recommend the acceptance of the article after considering the following concerns: 1. Please check the English again and typographical mistakes should be corrected. 2. Many figures in this manuscript should be re-depicted for clear representation. 3. Some important references of drug discovery from natural source may be added and cited in this manuscript. Such as “The application of in silico drug-likeness predictions in pharmaceutical research”

Author Response

Dear Reviewer,

We are greatly obliged for having received the Reviewers’ valuable opinion and helpful suggestion on our manuscript. All changes in the manuscript are marked in yellow and blue. The replies to the specific comments are listed below.

Please check the English again and typographical mistakes should be corrected.

Answer:The English language was corrected. Changes are marked in yellow in the text.

Many figures in this manuscript should be re-depicted for clear representation.

Answer:The figures have been corrected and re-attached

Some important references of drug discovery from natural source may be added and cited in this manuscript. Such as “The application of in silico drug-likeness predictions in pharmaceutical research”

Answer:Text and references have been supplemented, changes highlighted in blue

Please, accept my best regards,

Yours sincerely,

Agnieszka Szewczyk
